# Evaluating Perceptual Distance Models by Fitting Binomial Distributions to Two-Alternative Forced Choice Data

## Abstract

The two-alternative forced choice (2AFC) experimental method is popular in the visual perception literature, where practitioners aim to understand how human observers perceive distances within triplets made of a reference image and two distorted versions. In the past, this had been conducted in controlled environments, with triplets sharing images, so it was possible to rank the perceived quality. This ranking would then be used to evaluate perceptual distance models against the experimental data. Recently, crowd-sourced perceptual datasets have emerged, with no images shared between triplets, making ranking infeasible. Evaluating perceptual distance models using this data reduces the judgements on a triplet to a binary decision, namely, whether the distance model agrees with the human decision - which is suboptimal and prone to misleading conclusions. Instead, we statistically model the underlying decision-making process during 2AFC experiments using a binomial distribution. Having enough empirical data, we estimate a smooth and consistent distribution of the judgements on the reference-distorted distance plane, according to each distance model. By applying maximum likelihood, we estimate the parameter of the local binomial distribution, and a global measurement of the expected log-likelihood of the measured responses. We calculate meaningful and well-founded metrics for the distance model, beyond the mere prediction accuracy as percentage agreement, even with variable numbers of judgements per triplet – key advantages over both classical and neural network methods.

## 1 Introduction

The fundamental problem of replicating the human visual system's ability to compare images is increasingly important in computer vision and other machine learning tasks, especially with the advent of powerful image processing and generation algorithms. Many *perceptual distance models* have attempted to capture the behaviour of the human visual system, whether through extracting and comparing structures present in images (Wang et al., 2004; 2003), modelling mechanisms that are present in the visual pathway (Laparra et al., 2016; Hepburn et al., 2020), or more recently using neural networks pre-trained for classification to extract image features (Zhang et al., 2018; Ding et al., 2020).

The usual paradigm is to obtain a distance replicating human judgements on a dataset gathered through psychophysical experiments. In many datasets, those experiments consist of performing two-alternative forced choice (2AFC), where participants are presented with a reference image, and two distorted versions of that image (Fechner, 1948). The human observer is then asked to determine which distorted image is perceptually closer to the original. Sometimes these judgements are then transformed into a mean opinion score (MOS), which is then used to order the distorted images according to the perceived similarity to the corresponding reference (MOS from 0 to 100) (Ponomarenko et al., 2009; 2013; Sheikh et al., 2005; Larson & Chandler, 2010). Borrowing from chess, this uses the Swiss competition principle, and requires carefully selecting which images are shown to a given observer. Additionally, a distance model can be learned by assuming some distribution on the "internal score" a human observer assigns to a given image (Thurstone, 1994), and maximising the parameters of this distribution to fit the empirical data. This also requires triplets to have images

in common. Other methodologies include comparing a reference with only one distorted image and asking the participant how perceptible is the distortion (Lin et al., 2019).

Traditional perceptual datasets are collected in controlled environments, where variables such as the monitor, distance to screen and environmental lighting are managed. This results in a small but trustable set of measurements. Recently, practitioners have dropped those constraints in favour of gathering much greater amounts of data, using more reference images and a larger number of distortions. For example, the Berkeley Adobe Perceptual Patch Similarity (BAPPS) (Zhang et al., 2018) contains judgements performed on roughly 187,700 image patches using 425 distortions, compared to a dataset like TID2013 (Ponomarenko et al., 2013), with 25 reference images with 24 distortions applied. Additionally, BAPPS contains the raw perceptual judgements, as opposed to only the MOS, as it is infeasible to compute a ranking using triplets that do not share images (Tsukida et al., 2011). From this crowd sourced data, neural networks have been optimised to maximise the agreement with human observers in order to create a perceptual distance model (Zhang et al., 2018). However, evaluating perceptual distance models using such data is nontrivial, as a mapping from the distances, between the two distorted images and reference, to the proportion of people that judged the first distorted image closer to the reference, needs to be learnt. The CLIC dataset (Toderici et al., 2021) also contains the raw perceptual judgements, although with a varying number of judgements per triplet. Datasets containing only the MOS for each given reference-distorted image pair are also missing some key information that could aid in learning. In particular, the number of human participant judgements per image triplet from a 2AFC experiment can be used to infer uncertainty over the judgement.

Most work in the literature addresses the problem of estimating a distance (or measure of quality) between the images presented to the subjects from the binary responses to 2AFC tests. Many simplifications and added assumptions are necessary to make the problem tractable in experiments with different people with unknown response parameters governing their "internal scores" (Thurstone, 1994). However, it is important to note that, for the problem at hand, *we are not interested in estimating the perceived distance between test images presented to subjects*. Instead, we want to objectively evaluate how well different perceptual distance *models* explain the observed 2AFC data.

We follow relevant literature and model the perceptual judgements using a binomial distribution (Thurstone, 1994). We then compute the two distances between images within a triplet using a set of perceptual distance models. We then estimate the density of each score given these distances, including a degree of smoothing using Gaussian kernels, and locally fit a binomial model that maximises the likelihood of the judgements. Evaluation metrics such as the likelihood of a judgement according to the binomial model are simple to compute, and can explicitly account for a different number of human judgements used in the experiment, even for individual triplets. Fitting of this model requires little computation and can be easily parallelised. The paper is structured as follows: Sec. 2 gives an overview of the relevant literature. We will then present an overview of the proposed method in Sec. 3 and our findings using a selection of candidate distances in Sec. 4 evaluated on BAPPS and CLIC, also examining the robustness of the model with respect to hyperparameters.

## 2 RELATED WORK

### 2.1 PERCEPTUAL EXPERIMENTS

There are a number of methodologies that dictate how perceptual experiments are performed that differ in both the way the stimuli are presented, and the interaction the participant has with the experiment. We will focus on forced choice experiments, where the participant is asked to choose between several different stimuli. In image perception, the most commonly used experimental setup is two-alternative forced choice (2AFC). This is when the participant is presented with a reference image, and two distorted versions of the reference image, and usually asked to select the distorted image that is more similar to the reference. Popular datasets such as TID 2008 & 2013 (Ponomarenko et al., 2009; 2013) and BAPPS (Zhang et al., 2018) use this setup, however TID uses 2AFC experiments to *rank* distorted images. The mean opinion score (MOS) is then calculated based on this ranking for each reference-distorted pair, and only this is released. BAPPS however presents *random* triplets to observers and releases the unprocessed data, a proportion of human observers that preferred the second distorted image over the first. The CLIC dataset (Toderici et al., 2021) also releases the

2AFC data, however triplets have a variable number of judgements. An overview of the available visual perception datasets can be found in Appendix A.

Evaluating a perceptual distance model using the MOS is simple. One can simply compute the distance between each reference-distorted pair according to the model, and calculate the correlation between these distances and the MOS. However, with datasets like BAPPS, this is not possible. One evaluation metric is to force the decision to be binary: *according to the distance model, is the first distorted image closer to the reference than the second?* This ignores the number of judgements performed per triplet, wrongly equating a unanimous decision to one that is close to a tie.

Zhang et al. (2018) address this by transforming the two distances (reference-1st distorted, reference-2nd distorted) to the proportion of human observers that would judge the first distorted image closer to the reference. This is a non-linear transform that is parameterised by a simple neural network, and optimised to minimise the cross-entropy loss between the network predictions and the 2AFC experimental results. The network also allows the parametrisation of a perceptual distance model to be optimised to minimise the cross-entropy loss. However, this is not truly modelling the decision process when performing 2AFC experiments, and the evaluation consists of calculating the percentage of agreement between the network and human observer. Additionally, to compare different perceptual distances using BAPPS fairly, one would need to fit a separate neural network per distance, which quickly becomes computationally expensive.

Our proposed method assumes we have access to 2AFC experimental results, although it generalises to any alternative forced choice experiments such as the method of quadruples (Kingdom & Prins, 2010). We focus on BAPPS, as there is also no relationship between triplets (no ranking), which is what the proposed method is designed for. We also report results on the CLIC dataset, where there is a variable number of judgements per triplet.

## 2.2 PERCEPTUAL DISTANCES

In traditional perceptual literature, perceptual distances are hand-designed models, inspired by findings in vision science. Distances such as SSIM (Wang et al., 2004) and MS-SSIM (Wang et al., 2003) rely on the *structural similarity* between two images, i.e. how do humans perceive structure in an image. Separately, there are distances based on the *visibility of errors*; errors or differences between a reference and distorted image directly impact the similarity between the images. Models using this principle usually transform images to a more "perceptual" space and there compute a Euclidean distance. For example, in the normalised Laplacian pyramid distance (NLPD) (Laparra et al., 2016), such transformations are learned based on reducing redundancy in neighbouring pixels.

More recently, neural networks have been used to extract useful features for perceptual judgements. The Learned Perceptual Image Patch Similarity (LPIPS) metric (Zhang et al., 2018) uses networks pretrained for image classification to extract features, then learns a linear weighting of these features that correlates well with human perception. A neural network is then optimised to minimise the cross-entropy between the output of the network and 2AFC data. Although this successfully predicts perception, it lacks underlying assumptions on the psychophysical model that dictates human behaviour in 2AFC experiments. It is also unable to calculate the likelihood of a certain judgement. Deep Image Structure and Texture Similarity (DISTS) builds upon this to include a measure of texture similarity based on spatial averages (Ding et al., 2020). The Perceptual Information Metric (PIM) (Bhardwaj et al., 2020) finds a representation that maximises the mutual information between adjacent frames in videos. This representation is then used to compute distances.

For our candidate distances, which we wish to evaluate using the proposed method, we select a variety of traditional and deep learning-based metrics: Euclidean distance, NLPD (Laparra et al., 2016), SSIM (Wang et al., 2004), PIM (Bhardwaj et al., 2020), LPIPS (Zhang et al., 2018) and DISTS (Ding et al., 2020).

## 2.3 BAYESIAN APPROACH TO VISION

Bayesian approaches have been used throughout the perception literature. Maximum likelihood difference scaling (MLDS) (Maloney & Yang, 2003) models the observers ordering and fits a function to predict human perception in difference scales - the perceived difference between stimuli with a particular distortion (or particular direction in image space). With the addition of Gaussian noise to

simulate judgement uncertainty, the difference scales are optimised to maximise the likelihood of the human judgements belonging to the scales. This also allows the fitting of continuous functions defined by the practitioner, and can be used to estimate the difference scales in continuous physical measurements such as luminance and contrast.

Assuming that a stimuli's quality can be modelled as a Gaussian random variable, Thurstone proposed a number of different settings depending on the assumptions made on the underlying psychometrics (Thurstone, 1994). The means $\mu$ of these Gaussian random variables can be estimated by maximising the likelihood of a set of judgements belonging to a binomial distribution (Tsukida et al., 2011; Silverstein & Farrell, 2001; Jogan & Stocker, 2014), where the means relate to a *scale difference*, given an internal model within each participant.

Unlike the classical problem above, we do not need to rely on any assumed internal response model; rather we model the discrete decision process itself in the 2AFC tests. We apply the most generic and simple random model, the binomial distribution, having as a single parameter the probability of choosing one option vs. the other. This, together with the structure of our problem, which associates one candidate distance measurement to 2AFC tests, allows us to estimate the probability of choosing one distorted image over the other, for any number of experimental judgements, using the binomial distribution expression. As a result, we can reliably estimate the fitness of every candidate perceptual distance model in a conceptually and computationally simple way, and, more importantly, without adhering to model assumptions about the subjective responses of the subjects.

## 3 METHOD

Let us consider psychophysical data from the two-alternative forced choice (2AFC) between two degraded images and a reference image that are presented to the subjects, who must choose which one is visually closer to the reference image (Ponomarenko et al., 2009; 2013; Zhang et al., 2018; Toderici et al., 2021). We assume $T$ triplets of images, where each presented triplet $t$, $(\mathbf{x}_{ref}(t), \mathbf{x}_0(t), \mathbf{x}_1(t))$, receives a fixed number $M$ of responses. We define $n(t) \in [0, M]$ as the number of times $\mathbf{x}_1(t)$ is deemed to be closer to $\mathbf{x}_{ref}(t)$ than $\mathbf{x}_0(t)$ according to the observer. We propose that these choices are modelled by a binomial distribution $n \sim \mathcal{B}(M, P)$, where $P$ denotes the probability of $\mathbf{x}_1$ being chosen. The underlying hypothesis (the "distance hypothesis"), is that there is a perceptual distance model $d$ that maps two images to their corresponding visual distance, such that the choice count $n$ is conditionally independent of $\{\mathbf{x}_{ref}, \mathbf{x}_0, \mathbf{x}_1\}$ given $(d(\mathbf{x}_{ref}, \mathbf{x}_0), d(\mathbf{x}_{ref}, \mathbf{x}_1))$. We also define the distance between the reference and first distorted as $d_0(t) = d(\mathbf{x}_{ref}(t), \mathbf{x}_0(t))$ and distance between the reference and second distorted as $d_1(t) = d(\mathbf{x}_{ref}(t), \mathbf{x}_1(t))$. Under the assumed binomial choice model, the previous hypothesis is equivalent to stating that there is a function $f$ such that the binomial parameter $P$ controlling the probability of the choice is:

$$P(\mathbf{x}_{ref}, \mathbf{x}_0, \mathbf{x}_1) = f(d_0, d_1).$$

We will refer to the function $f$ as $P(d_0, d_1)$ for simplicity, and $P(d_0, d_1)$ corresponding to a particular triplet as $P(d_0(t), d_1(t))$. Given a set of psychophysical data corresponding to the previous experimental setup and a perceptual distance model $d$, here we address the problem of measuring how well that candidate function fits the empirical data under the assumed binomial process. At this point it is important to note that we should not choose $P(d_0(t), d_1(t))$ simply as its most likely value $(n(t)/M)$ according to that particular triplet $t$. Instead, because there is a random process involved, we must consider other triplet results in the neighbourhood of that $(d_0(t), d_1(t))$ location to average the local results and obtain a smooth and consistent $P(d_0, d_1)$ function on the whole $(d_0, d_1)$ distance plane. The reliability of $P(d_0, d_1)$ critically depends on having a large set of results providing a dense sampling of the $(d_0, d_1)$ plane.

### 3.1 MARGINAL UNIFORMISATION

The most common approach to estimate a smooth function from a set of discrete events is to use adaptive kernels, with sizes depending on the density of values at a given point when estimating the density. Alternatively, we can search for transformations relocating the samples such that they cover the space fully and as evenly as possible. In this work, for each candidate distance, we transform the pairs of distances $\{d_0, d_1\}$, such that the resulting set of points is marginally uniform in the range

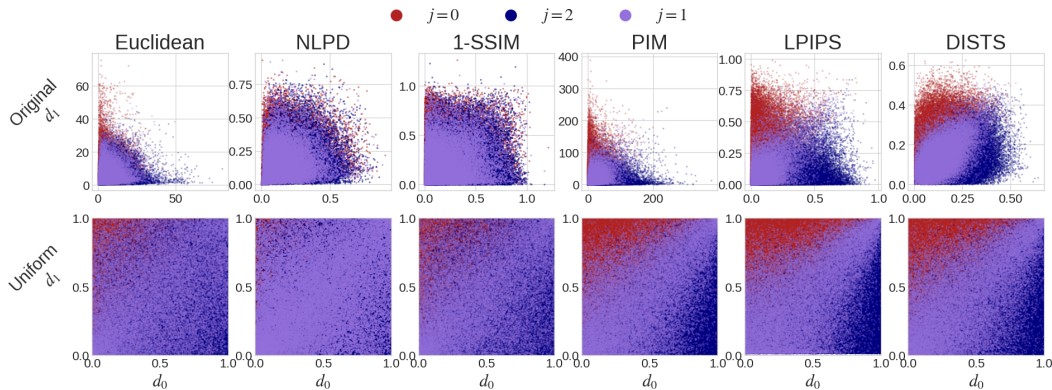

Figure 1: Scatter plot of candidate distances in their original space (top row) and uniformised (bottom row). Shown are the training samples from the BAPPS dataset and the colour indicates the judgement assigned to triplet according to 2 observers.

$[0, 1]$. The marginal uniformisation transform is given by the histogram-equalisation solution

$$u = U_{(k)}^i(x_i^{(k)}) = \int_{-\infty}^{x_i^{(k)}} p_i(x'^{(k)}_i) dx'^{(k)}_i. \tag{1}$$

where $x_i^{(k)}$ is a sample in the $k$th marginal and $p_i$ is approximated with histograms, for which computing the cumulative density function is straightforward (Laparra et al., 2011). In doing so we are transforming our data to a (fairly) uniform domain. This non-linear transformation facilitates the numerical density estimations from the discrete data but it is transparent for the posterior computations. Fig.1 shows some examples of distances transformed to a uniform domain.

## 3.2 MAXIMUM-LIKELIHOOD MODEL-FITTING OF THE EMPIRICAL DATA

We estimate the empirical densities $\{p(j, d_0, d_1), j = 1 \ldots M\}$, which model how likely are the events $\{(d_0, d_1), n = j\}$. As explained before, to obtain a continuous probability density function (PDF) from a set of discrete events, we use a Gaussian kernel integrating the corresponding samples on the distance plane. This is done separately for each $j$, ensuring that $\int p(j, d_0, d_1), \mathrm{d}d_0 \, \mathrm{d}d_1 = P(j) = 1/T \sum \delta(n(t) - j)$. Then we compute the overall density on the distance place as $p(d_0, d_1) = \sum_{j=1}^{M} p(j, d_0, d_1)$. (Note that we can take advantage of the symmetry $p(d_0, d_1) = p(d_1, d_0)$, by enforcing it.) Now we can estimate the conditional probabilities

$$Pr(n = j | d_0, d_1) = \frac{p(j, d_0, d_1)}{p(d_0, d_1)}, j = 1 \ldots M. \tag{2}$$

In practice, to obtain a smooth PDF estimation from a set of discrete points on the $(d_0, d_1)$ plane, we first need to define a grid made up of evenly spaced points, for which we will estimate the conditional probabilities. Gaussian kernels of a suitable size, centered at each of the discrete points in the training set and evaluated on the grid, are then summed. This is applied separately to each of the sets $\{(d_0, d_1), n = j\}$, thus obtaining the estimations $p(j, d_0, d_1)$, for all $j$s. Finally, we compute their sum $(p(d_0, d_1))$ and apply Eq. 2 to obtain the estimated conditional probabilities, $Pr(n = j | d_0, d_1)$. On the other hand, according to the assumed model, $n(t) \sim \mathcal{B}(M, P(d_0(t), d_1(t)))$, we know the theoretical binomial probability of each $n$ as a function of $P(d_0, d_1)$ and $M$, that we term $Pr_{\mathcal{B}}(n = j; M, P(d_0, d_1))$. From the above, we estimate $\hat{P}(d_0, d_1)$, the probability parameter maximising the log-likelihood of the observations for every $(d_0, d_1)$:

$$\hat{P}(d_0, d_1) = \arg \max_P L(d_0, d_1; M, P), \tag{3}$$

where

$$L(d_0, d_1; M, P) = \sum_{j=1}^{M} Pr(n = j | d_0, d_1) \log Pr_{\mathcal{B}}(n = j; M, P) \tag{4}$$

is the likelihood of the obtained 2AFC answers having been generated from a binomial distribution with parameters $P(d_0, d_1)$ and $M$. We have that:

$$Pr_{\mathcal{B}}(n = j; M, P) = \frac{M!}{j!(M-j)!} P^j (1-P)^{M-j}, \tag{5}$$

We use the latter expression in Eq. 4 for choosing the $P$ that maximises the likelihood of the empirical probabilities $\{Pr(n = j|d_0, d_1)\}$:

$$\hat{P}(d_0, d_1) = \arg\max_P \sum_{j=1}^{M} Pr(n = j|d_0, d_1) \begin{bmatrix} \log\left(\frac{M!}{j!(M-j)!}\right) + j \log P \\ + (M-j)\log(1-P) \end{bmatrix}$$

$$= \frac{1}{M} \sum_{j=1}^{M} j \cdot Pr(n = j|d_0, d_1), \tag{6}$$

where the last equality is obtained by differentiating the argument of the right-hand side in the first equation w.r.t. $P$, equating to 0 and solving for $P$.

It is important to note that the modelling of the decision-making process does not depend on $M$, but contributes to the empirical data and informs us the number of votes per triplets, allowing us to evaluate with a variable number of judgements with the same underlying learned distribution.

Finally, we average the local (in $(d_0, d_1)$) optimal log-likelihood $\hat{L}(d_0, d_1) = L(d_0, d_1; M, \hat{P}(d_0, d_1))$ on the whole $(d_0, d_1)$ plane providing a global evaluation of how well the binomial distribution with the $d$ distance fits the empirical 2AFC data. Fig .2 shows an example of the method, using a small amount of data points.

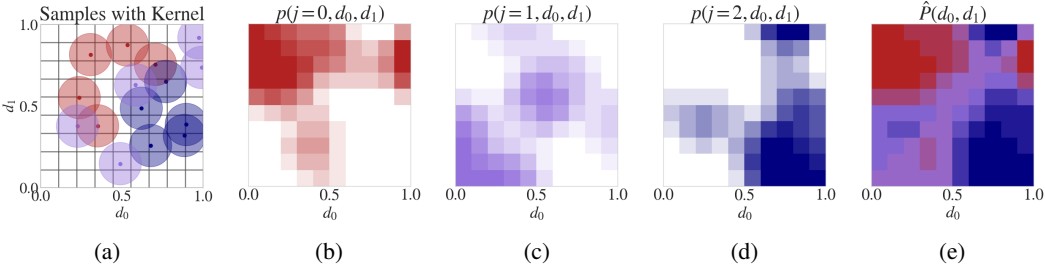

|   |   |   |   |   |
|---|---|---|---|---|
| (a) | (b) | (c) | (d) | (e) |

Figure 2: An example using 5 data points, $M = 2$ for each $j = \{0, 1, 2\}$. (a) Samples with a Gaussian kernel applied (the circle represents the standard deviation) and the $10 \times 10$ grid for which we estimate Eq. 2. (b), (c) and (d) are the estimated conditional distributions for each value of $j$, and (e) is the distribution after maximum likelihood estimation according to Eq. 6.

We note that (Zhang et al., 2018) used a neural network to estimate $P(d_0, d_1)$. In contrast, we use a single model for both estimating $P(d_0, d_1)$ and evaluating the perceptual distance model, which is arguably more consistent, and requires orders of magnitude less training time and parameters.

## 3.3 EVALUATION METRICS

We propose several evaluation metrics to compare the fitting of the binomial to each distance model. Two types of evaluation metrics will be used; (1) based on the agreement of the output $\hat{P}(d_0, d_1)$, and (2) based on the log-likelihood of judgements according to the learned binomial distribution.

If we apply the criterion of comparing the most likely outcome of the binomial distribution $\mathcal{B}(M, P)$, which is $\lfloor (M+1)P \rfloor$ (Feller, 1991), with the actual judgement $n(t)$, both normalised to $M$ for each triplet, we obtain a percentage agreement between our probability model $\mathcal{B}(M, \hat{P}(d_0(t), d_1(t)))$ and the human judgements $n(t)$, yielding the following expression:

$$\text{AJ}(\{n(t), t = 1 \ldots T\}, \hat{P}, M) = 100 - \frac{100}{T} \sum_{t=1}^{T} \left| \frac{\lfloor (M+1)\hat{P}(d_0(t), d_1(t)) \rfloor - n(t)}{M} \right|. \tag{7}$$

Note that the $M$ used here corresponds to the judgement $n(t)$, using the already fit probability model $\hat{P}$. The $M$ used in the evaluation can be different from the one used in the estimation of $\hat{P}$, for example, when a test set contains a different number of judgements.

To maximise the agreement for the learned model, one can generate random samples $\hat{n}(t)$ from our learned distribution $\hat{n}(t) \sim \mathcal{B}(M, \hat{P}(d_0(t), d_1(t)))$, and see the agreement between the empirical and the simulated judgements, according to our learned model. This provides a reference for the case that the observed judgements follow the fitted binomial model exactly.

Due to our assumptions of an underlying probability model for the decision process, we can also evaluate log-likelihoods of judgements according to our learned model. Rather than just measure the percentage of judgements that agree with our model, we can evaluate the negative log-likelihood of the empirical data according to the learned binomial model given by

$$\text{NLL}(\{n(t), t = 1 \dots T\}, \hat{P}, M) = -\frac{1}{T} \sum_{t=1}^{T} \log Pr_{\mathcal{B}}(n(t); M, \hat{P}(d_0(t), d_1(t))) \tag{8}$$

Similarly to the agreement of judgements, we can sample from our learned distribution with $M$ judgements $\hat{n}(t) \sim \mathcal{B}(M, \hat{P}(d_0(t), d_1(t)))$ and evaluate the negative log-likelihoods of these judgements, to get a minimum possible negative log-likelihood achieved by a distance, which is an estimate of the entropy of the learned binomial distribution. This would mean using $\hat{n}(t)$ rather than using $n(t)$ from the dataset, giving $\text{NLL}(\hat{n}, \hat{P}, M)$.

Finally, a regularly used metric is the agreement purely between the decisions; *does the perceptual distance select the same distorted image as the humans?* For a set of experiments, this is given by

$$2\text{AFC} = \frac{1}{T} \sum_{t=1}^{T} \hat{p}(d_0(t), d_1(t)) \cdot \frac{n(t)}{M} + (1 - \frac{n(t)}{M}) \cdot (1 - \hat{p}(d_0(t), d_1(t))), \tag{9}$$

where $n(t)$ is the number of humans that selected the first distorted image as closer to the reference, and $\hat{p} \in \{0, 1\}$ is the preference of the perceptual distance model i.e. if $d_0 > d_1$.

### 3.4 Extension to different number of judgements

We can generalise the result from the previous section to a different number of judgements $M$ per triplet $t$, $M_t$, by transforming the $M_t$ judgements of each given triplet into binary judgements ($M = 1$) on $M_t$ identical triplets. Now we can construct the conditional distributions $p(j = \{0, 1\}, d_0, d_1)$, where the $P$ that maximises the likelihood of empirical probabilities is simply $\hat{P}(d_0, d_1) = p(j = 1, d_0, d_1)/p(d_0, d_1)$. For example, where $M = 2$ and both participants select $\mathbf{x_0}$ ($n(t) = 2$), this is equivalent to two individual judgements preferring $\mathbf{x_0}$ in two identical triplets. The evaluation metrics remain consistent, but using $M_t$ for each judgement rather than $M$.

## 4 Experiments

We can apply our likelihood model to existing perceptual distances, using maximum likelihood to fit a probability model to the pairs of distances $\{d_0, d_1\}$. We do so for 6 candidate distances, and use the training set of the BAPPS dataset, containing triplets in the form $\{\mathbf{x}_{ref}, \mathbf{x}_0, \mathbf{x}_1\}$ which are patches of size $64 \times 64$. The training set contains $M = 2$ judgements, so the possible values of preference $n(t)/M$ are limited to the set $\{0.0, 0.5, 1.0\}$, where $0.0$ means that both observers deem $\mathbf{x}_0$ to be closer to the reference image than $\mathbf{x}_1$. We evaluate our model fit on the training data on the BAPPS validation set, which contains $M = 5$ judgements. Note that, given we are directly estimating $P(d_0, d_1)$, we can train and evaluate on a different number of $M$ judgements. To obtain a smooth PDF from the set of discrete $\{d_0, d_1\}$ samples, we use a Gaussian kernel with a constant width. This is set as a multiple of the range of the data, which when uniformised is in $[0, 1]$. The width $\sigma$ can be set by the user – in our experiments, we found that the method was robust to different $\sigma$ values, but depended on the amount of data that covers the plane, i.e., less data requires a larger $\sigma$. For the BAPPS dataset, we set $\sigma = \frac{1}{44}$. We use a $100 \times 100$ grid to estimate the conditionals in Eq. 2. Sec. 4.3 examines the proposed method's robustness to changes in these hyperparameters.

Table 1: Evaluation metrics on the BAPPS validation set ($M = 5$). Lower NLL is better.

| Measure | Euclidean | NLPD | SSIM | PIM | LPIPS | DISTS |
|---|---|---|---|---|---|---|
| $AJ(n, \hat{P}, M)$ (%)↑ | 68.615 | 68.049 | 70.470 | 80.226 | 80.290 | 79.325 |
| $AJ(\hat{n}, \hat{P}, M)$ (%)↑ | 74.197 | 73.682 | 74.074 | 79.344 | 80.126 | 79.102 |
| $NLL(n, \hat{P}, M)$↓ | 1.058 | 1.074 | 1.029 | 0.775 | 0.761 | 0.798 |
| $NLL(\hat{n}, \hat{P}, M)$↓ | 0.962 | 0.969 | 0.948 | 0.797 | 0.782 | 0.810 |
| 2AFC Score↑ | 0.668 | 0.665 | 0.688 | 0.776 | 0.770 | 0.766 |

Table 2: 2AFC Scores (Eq. 9) on the BAPPS test set, where distance only is checking if $d_0 > d_1$, and $\hat{P}(d_0, d_1)$ is our approach. *LPIPS has been optimised to replicate decisions in the BAPPS training set using a neural network mapping $(d_0, d_1)$ to $n(t)/M$.

| 2AFC Score↑ | Euclidean | NLPD | SSIM | PIM | LPIPS* | DISTS |
|---|---|---|---|---|---|---|
| Distance Only | 0.634 | 0.627 | 0.631 | 0.693 | 0.694 | 0.685 |
| $\hat{P}(d_0, d_1)$ | 0.629 | 0.629 | 0.632 | 0.697 | 0.689 | 0.686 |

## 4.1 RESULTS

We evaluate the metrics in Sec. 3.3 for the test set of BAPPS. Table 1 contains the results after optimising separate binomial models for each of our six distance models. The first three models (Euclidean, NLPD and SSIM) achieve similar results in agreement measures and negative log-likelihoods. The three deep-learning-based models (PIM, LPIPS and DISTS) achieve superior performance. Additionally, the negative log-likelihoods for the simulated judgements $NLL(\hat{n}(t), \hat{P}, M)$, the minimum possible, is lower (better), as expected. Evaluation metrics on the training set can be see in Appendix B. Table 2 compares the 2AFC score using just the raw distance, vs using $\hat{P}(d_0, d_1)$ (as in Table 1). Most distances see just a slight increase in score, apart from Euclidean and LPIPS. Note that LPIPS parameters have been optimised to explicitly minimise the mean squared error between the network outputs and the ground truth proportion $n(t)/M$.

In addition, it is highly informative to visualise $\hat{P}(d_0, d_1)$ for different distances, to see the separation or amount of uncertainty along the diagonal, where $d_0$ and $d_1$ are similar. Fig. 3 shows $\hat{P}(d_0, d_1)$, revealing that the three deep-learning-based metrics offer more certainty in the top left ($d_0 << d_1$) and bottom right ($d_0 >> d_1$), whereas the other distances display a wider area around $0.3 < \hat{P}(d_0, d_1) < 0.7$, indicating higher uncertainty on the decision. Additionally, for the DL-based metrics, as the distances $\{d_0, d_1\}$ increase (top right), the uncertainty region grows smaller, reflecting how humans perceive sharper differences in images far from the reference.

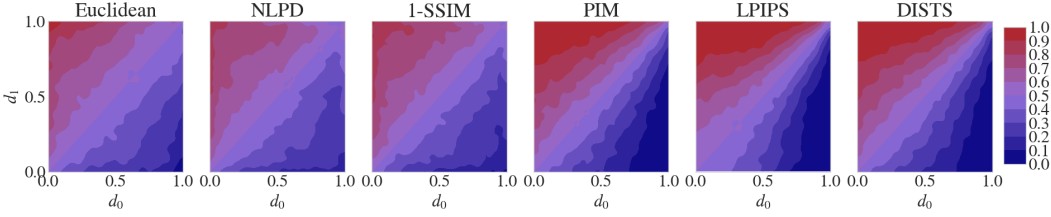

Figure 3: $\hat{P}(d_0, d_1)$ estimated from the BAPPS training set for different perceptual distance models.

## 4.2 INTERPRETABILITY

One advantage of explicitly modelling the decision is that we can evaluate the negative log-likelihood of different $j$ values using Eq. 8, allowing users to query the model for a given perceptual distance model. Fig 4 shows the negative log-likelihood of a different number of observers $j = [0, 5]$ preferring the far right image over the middle. (See other examples Appendix C). Practitioners are

able to see where a certain perceptual distance model fails, evaluate the likelihood of the decision, and how the likelihood will change with changes in $(d_0, d_1)$ and the number of judgements $M$.

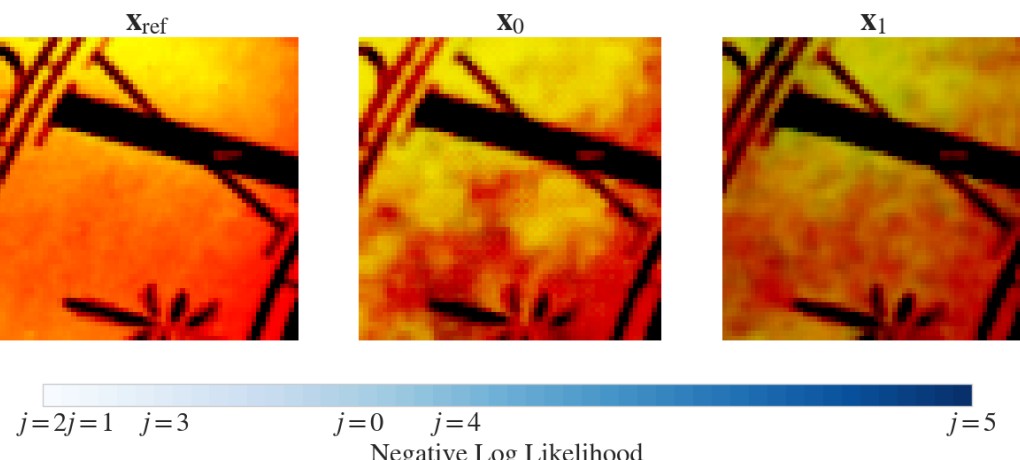

Figure 4: An example of querying the probability model for different number of people that prefer $\mathbf{x}_0$ over $\mathbf{x_1}$, $j = [0, 5]$, values for a triplet from the BAPPS test set, according to DISTS. Shown is negative log-likelihood, where white is more likely and blue is less likely.

### 4.3 ROBUSTNESS

There are two main hyperparameters of interest; the width of the Gaussian window $\sigma$ for smoothing, and the resolution of the grid used to estimate $\hat{P}(d_0, d_1)$. We find the optimal $\hat{P}(d_0, d_1)$ for the training set of BAPPS ($M = 2$) by continuously varying the chosen hyperparameter, and plot the negative log-likelihood and percentage agreement when evaluating on the test set ($M = 5$).

The kernel width sets the amount of smoothing applied to our point-wise estimates from the training set. We test a wide range of values $\sigma \in [0.0037, 1]$ (where the data are in $[0, 1]$), and fix the grid to $100 \times 100$. Fig. 5a shows that a large $\sigma$ (heavy smoothing) results in performance loss, with a large flat section for $\sigma < 0.1$ that achieves similar performance. The ability to visualise the decision surface (Fig 3) allows the user to decide a $\sigma$ that determines the smoothness of the estimated distribution $\hat{P}(d_0, d_1)$.

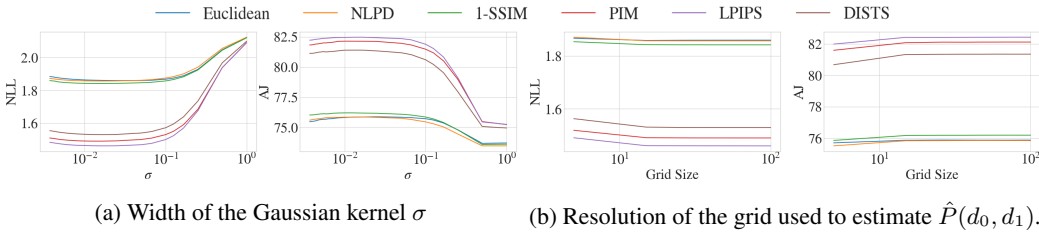

(a) Width of the Gaussian kernel $\sigma$      (b) Resolution of the grid used to estimate $\hat{P}(d_0, d_1)$.

Figure 5: Robustness of the method with relation to (left) the width of the Gaussian kernel $\sigma$ using to estimate the conditionals and (right) the resolution of the grid used to estimate $\hat{P}(d_0, d_1)$. In each subplot, left is the negative log-likelihood of the judgements belonging to the learned model (Eq. 6), right is the agreements of judgements (Eq. 7).

The grid size sets the resolution of $\hat{P}(d_0, d_1)$. Fig. 2 (a) shows an example of a $10 \times 10$ grid on the data in $[0, 1]$. We test grid sizes in the range $[5, 100]$. Fig.5b shows that a too coarse grid does not allow proper estimation, and more partitions than roughly 20 see no gain in performance. The lower the grid size, the less parameters and the less computation needed to construct the estimation. This also depends on the spread of data in $\{d_0, d_1\}$ after marginal uniformisation, but for data similarly distributed to BAPPS, one would expect a stable performance across grid sizes.

## 4.4 DIFFERENT NUMBER OF JUDGEMENTS

We also present results on the Challenge on Learned Image Compression (CLIC) 2021 dataset, which contains $t$ triplets with different numbers of judgements $M_t$. Judgements which used an "anchor", where one of the distorted images is actually the reference image, were removed as including these images interacts with the uniformisation transform as they are on the edge of the support ($d_0$ or $d_1 = 0.0$). We use the oracle set to optimise, and the validation set to evaluate. The oracle set contains 119,901 triplets with $M_t = \{1, 2\}$ and the validation set contains 4807 triplets with $M_t = [1, 10]$, although a large percentage have one judgement. The distribution of $M_t$ can be seen in Appendix D.

The results on the CLIC dataset are similar, as seen in Table 3, with Euclidean, NLPD, and SSIM struggling to distinguish between distorted images. This is due to the non-uniformity in the CLIC measurements, despite the marginal uniformisation. This can be seen in Appendix D, where these three distances display a large amount of uncertainty in the surface. More data could alleviate this issue, in order to estimate a smooth and consistent $\hat{P}(d_0, d_1)$. PIM, LPIPS, and DISTS display much more expected behaviour, with PIM achieving a 2AFC score on the test set of $0.7305$.

Table 3: Results on the CLIC validation set ($M_t = [1, 10]$). Lower NLL is better.

| Measure | Euclidean | NLPD | SSIM | PIM | LPIPS | DISTS |
|---|---|---|---|---|---|---|
| AJ$(n, \hat{P}, M)$ (%)↑ | 53.185 | 51.860 | 56.315 | 73.034 | 73.226 | 72.082 |
| AJ$(\hat{n}, \hat{P}, M)$ (%)↑ | 51.439 | 50.996 | 52.204 | 69.830 | 69.698 | 68.999 |
| NLL$(n, \hat{P}, M)$↓ | 0.694 | 0.694 | 0.689 | 0.548 | 0.543 | 0.556 |
| NLL$(\hat{n}, \hat{P}, M)$↓ | 0.694 | 0.695 | 0.693 | 0.567 | 0.567 | 0.582 |
| 2AFC Score↑ | 0.5304 | 0.5173 | 0.5619 | 0.7297 | 0.7317 | 0.7202 |

## 5 CONCLUSION

We present a method for evaluating the ability of perceptual distance models to explain two-alternative forced choice experimental data, using a simple assumption about the observer's decision-making process. We rely on kernel integration and marginal uniformisation to compute a smooth and consistent estimate of the involved PDFs on the 2-distance plane $(d_0, d_1)$, where maximum likelihood is applied to estimate the choice weight given a pair of distance values from the perceptual model. We apply it to the BAPPS dataset, obtaining a similar ranking for the visual models as in previous works but with additional insights from the richer pool of evaluation metrics. The method is robust to changes in the values of its two hyperparameters, allowing practitioners to quickly and reliably evaluate perceptual distance models without needing additional training or tuning.

The main limitations of the proposed method comes from the way the data is gathered. Traditional perceptual datasets, such as TID (Ponomarenko et al., 2009; 2013) use a ranking algorithm to decide which triplets to show the observer. For those data sets our method does not take advantage of this information, as it assumes the triplets are randomly selected. In addition, as the ranking method is expensive in terms of the required number of judgements and does not scale well to large amounts of data, the number of triplets tends to be low. In contrast, our method requires a large number of samples to ensure reliable PDF estimations on $(d_0, d_1)$, which is consistent with more recent crowd-sourced datasets.

Regarding possible extensions, it is feasible to parametrise a distance model and optimise it for maximising the log-likelihood. This could lead to metrics like LPIPS, where the distance model is optimised based on the training data. Another line is to analyse the (subtle) practical impact of the mismatch between the numbers of judgements $M$ in training and testing. Note that, although $\hat{P}(d_0, d_1)$ does not depend on $M$, the data used to estimate the PDF does.

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
