## A    PERCEPTUAL DATASETS

Table 4 shows the existing perceptual datasets, and what sort of experimental setup was used. Whilst most datasets use the 2AFC setup, some datasets (TID, CSIQ, CLIC) use an Elo ranking system to decide which images to show a particular observer. This results in a dataset where each triplet judgement is not independent, and results in triplets having a different number of judgements $M$.

BAPPS and CLIC are the only datasets that release the raw 2AFC ratings, but they differ in that BAPPS ensures the same number of judgements for each triplet, and each observer is shown random triplets. This is the setting that the proposed method was designed for, but we can still apply it to others.

Table 4: Detailed description of existing perceptual datasets: TID 2008(Ponomarenko et al., 2009), TID 2013(Ponomarenko et al., 2013), CSIQ (Larson & Chandler, 2010), LIVE (Sheikh et al., 2005), BAPPS (Zhang et al., 2018) and CLIC (Toderici et al., 2021).

| Dataset | Method | Image Sizes | No. of Images | No. of Distortions | No. of Triplets | Total No. of Judgements | Type of Judgement Released |
|---|---|---|---|---|---|---|---|
| TID 2008 | 2AFC sorting | 512x384 | 25 | 17 | 2k | 256k | MOS |
| TID 2013 | 2AFC sorting | 512x384 | 25 | 24 | 3k | 5k | MOS |
| CSIQ | 2AFC | 512x512 | 30 | 6 | 866 | 5k | DMOS |
| LIVE | 5 level scale | 768x512 | 29 | 5 | 779 | 25k | DMOS |
| BAPPS Train | 2AFC | 64x64 | 151k | 425 | 151k | 302k | 2AFC |
| BAPPS JND | JND | 64x64 | 10k | 425 | 10k | 29k | True/False |
| BAPPS Validation | 2AFC | 64x64 | 36k | 425+ | 36k | 182k | 2AFC |
| CLIC 2021 | 2AFC | 768x768 | 315 | | 119k | 120k | 2AFC |

## B    BAPPS ADDITIONAL RESULTS

Here we present additional results on the training and test validation sets of BAPPS. We also separately report evaluation metrics per distortion used in the BAPPS test set for a more in-depth comparison of metrics.

Table 5 shows evaluation metrics on both the training and test set of BAPPS. The training set has been used to fit $\hat{P}(d_0, d_1)$. We see a consistent behaviour across sets, despite the different number of judgements $M$ for the train and test sets.

Table 6 shows a breakdown of the agreement of judgements (AJ) Eq. 7, negative log-likelihood (NLL) Eq. 8 and 2AFC score Eq. 9, evaluated on the test set of BAPPS. We split the dataset into the category of distortion used, namely: Traditional (4720 triplets), CNN (4720 triplets), Color (9440 triplets), Deblur (1888 triplets), Frame interpolation (10856) and Super resolution (4720 triplets). Details on these distortions can be found in Zhang et al. (2018).

## C    INTERPRETABILITY - MORE EXAMPLES

The negative log-likelihood in Eq 8 depends on $\hat{P}(d_0, d_1)$, and in order to visualise this, here we present several examples of evaluating the NLL of different $j = [0, 5]$. We use triplets where one distorted image is extremely close to the reference and the decision is clear (Fig 6), one distorted image is extremely far from the original and the decision is clear (Fig 7) and finally a triplet where the decision is borderline as both distorted images are far from the original (Fig 8).

Table 5: Results on the BAPPS dataset (Zhang et al., 2018). In the training set, there are 2 judgements per triplet ($M = 2$) and in the test set, 5 ($M = 5$). Lower NLL is better.

| Measure | | Euclidean | NLPD | SSIM | PIM | LPIPS | DISTS |
|---|---|---|---|---|---|---|---|
| AJ$(n, \hat{P}, M)$ (%)↑ | Train | 68.615 | 68.049 | 70.470 | 80.226 | 80.290 | 79.325 |
| | Test | 75.345 | 75.195 | 75.653 | 81.598 | 82.048 | 80.841 |
| AJ$(\hat{n}, \hat{P}, M)$ (%)↑ | Train | 74.197 | 73.682 | 74.074 | 79.344 | 80.126 | 79.102 |
| | Test | 82.650 | 82.544 | 82.795 | 84.216 | 84.461 | 84.179 |
| NLL$(n, \hat{P}, M)$↓ | Train | 1.058 | 1.074 | 1.029 | 0.775 | 0.761 | 0.798 |
| | Test | 1.889 | 1.885 | 1.867 | 1.522 | 1.490 | 1.557 |
| NLL$(\hat{n}, \hat{P}, M)$↓ | Train | 0.962 | 0.969 | 0.948 | 0.797 | 0.782 | 0.810 |
| | Test | 1.480 | 1.491 | 1.476 | 1.376 | 1.361 | 1.386 |
| 2AFC Score↑ | Train | 0.6675 | 0.6648 | 0.6880 | 0.7763 | 0.7698 | 0.7661 |
| | Test | 0.6289 | 0.6287 | 0.6319 | 0.6971 | 0.6890 | 0.6862 |

Table 6: Evaluation metrics on the BAPPS validation set, split by distortion applied.

| Distance | Measure | Distortion | | | | | |
|---|---|---|---|---|---|---|---|
| | | Traditional | CNN | Color | Deblur | Frame Interp. | Super Resolution |
| Euclidean | AJ$(n, \hat{P}, M)$ (%)↑ | 65.458 | 77.784 | 78.352 | 78.256 | 75.932 | 76.415 |
| | NLL$(n, \hat{P}, M)$↓ | 2.530 | 1.713 | 1.732 | 1.715 | 1.826 | 1.820 |
| | 2AFC Score↑ | 0.554 | 0.807 | 0.622 | 0.579 | 0.564 | 0.664 |
| NLPD | AJ$(n, \hat{P}, M)$ (%)↑ | 66.585 | 76.805 | 76.047 | 78.318 | 76.250 | 77.139 |
| | NLL$(n, \hat{P}, M)$↓ | 2.470 | 1.759 | 1.843 | 1.704 | 1.816 | 1.779 |
| | 2AFC Score↑ | 0.578 | 0.802 | 0.592 | 0.576 | 0.559 | 0.670 |
| SSIM | AJ$(n, \hat{P}, M)$ (%)↑ | 67.992 | 78.903 | 76.068 | 78.430 | 77.373 | 76.476 |
| | NLL$(n, \hat{P}, M)$↓ | 2.402 | 1.638 | 1.850 | 1.700 | 1.752 | 1.823 |
| | 2AFC Score↑ | 0.605 | 0.808 | 0.602 | 0.586 | 0.572 | 0.651 |
| PIM | AJ$(n, \hat{P}, M)$ (%)↑ | 81.428 | 87.013 | 80.233 | 81.422 | 81.981 | 81.780 |
| | NLL$(n, \hat{P}, M)$↓ | 1.541 | 1.150 | 1.619 | 1.551 | 1.492 | 1.504 |
| | 2AFC Score↑ | 0.767 | 0.838 | 0.652 | 0.622 | 0.632 | 0.716 |
| LPIPS | AJ$(n, \hat{P}, M)$ (%)↑ | 80.585 | 88.136 | 80.869 | 81.036 | 81.589 | 82.824 |
| | NLL$(n, \hat{P}, M)$↓ | 1.596 | 1.059 | 1.564 | 1.556 | 1.515 | 1.436 |
| | 2AFC Score↑ | 0.748 | 0.837 | 0.655 | 0.614 | 0.587 | 0.699 |
| DISTS | AJ$(n, \hat{P}, M)$ (%)↑ | 80.364 | 85.996 | 79.186 | 80.152 | 81.261 | 81.787 |
| | NLL$(n, \hat{P}, M)$↓ | 1.609 | 1.196 | 1.656 | 1.613 | 1.535 | 1.511 |
| | 2AFC Score↑ | 0.757 | 0.832 | 0.639 | 0.602 | 0.626 | 0.714 |

# D CLIC

We include additional information regarding the CLIC dataset used, including the distribution of the number of judgements $M_t$ per triplet. We also show additional visualisations of the triplets in the $(d_0, d_1)$ before and after the uniformistaion transformation, as well as evaluation metrics using both the training and test set.

## D.1 DISTRIBUTION OF NUMBER OF JUDGEMENTS

The CLIC 2021 subset we use to train (the oracle set) consists of 119,901 triplets with the number of judgements $M_t = \{1, 2\}$ and results of the judgements $j$, where the distribution can be seen in Fig. 11. We also show the distributio of $j$ for each $M_t$. Most of the triplets have one judgement, with roughly uniform $j = \{0, 1\}$. For the triplets with 2 judgements, the majority are indecisive with $j = 1$.

The same distribution for the subset used for evaluation (the validation set) with $M_t = [1, 10]$ can be seen in Fig 10. The vast majority of triplets also contain only 1 judgement, where the distribution

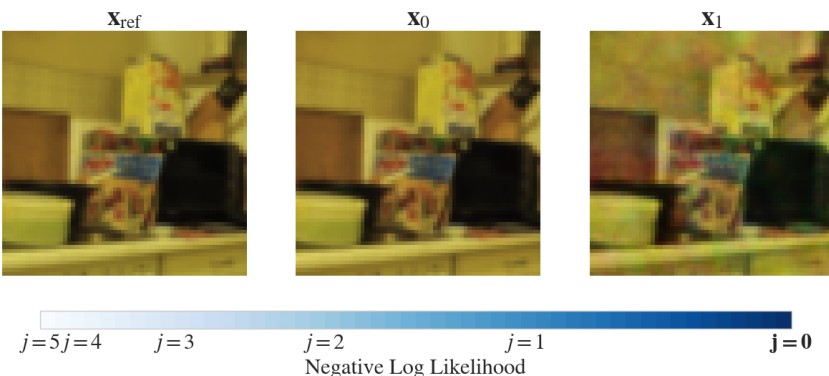

Figure 6: Example of valuating the negative log-likelihood $j = [0, 5]$ according to DISTS for a triplet from the BAPPS test set where one image $\mathbf{x}_0$ is close to the reference $\mathbf{x}_{\text{ref}}$. White is more likely and blue is less likely.

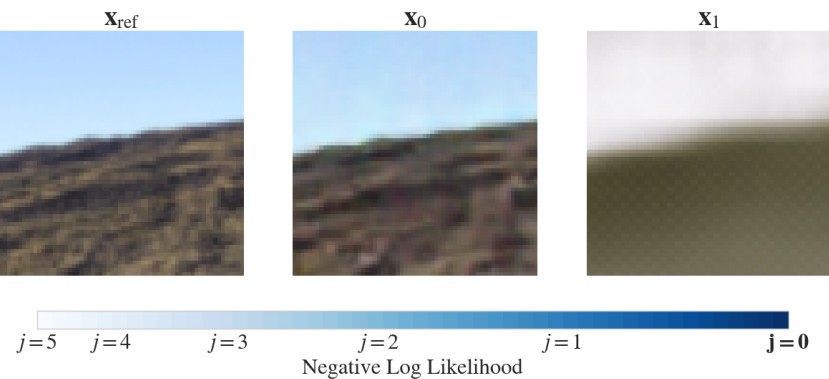

Figure 7: Example of valuating the negative log-likelihood $j = [0, 5]$ according to DISTS for a triplet from the BAPPS test set where one image $\mathbf{x}_0$ is far from the reference $\mathbf{x}_{\text{ref}}$. White is more likely and blue is less likely.

of $j$ similar to that of the training set. The set also includes a small number of triplets with more judgements, varying in distribution of $j$.

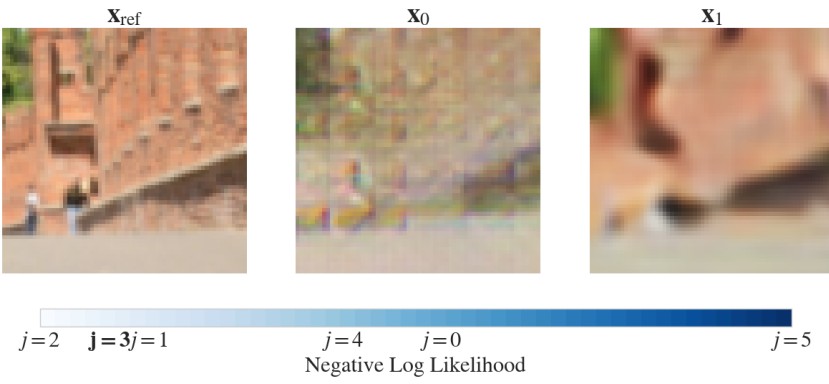

Figure 8: Example of valuating the negative log-likelihood $j = [0, 5]$ according to DISTS for a triplet from the BAPPS test set where both images $\{\mathbf{x}_0, \mathbf{x}_1\}$ are far from the reference $\mathbf{x}_{\text{ref}}$. White is more likely and blue is less likely.

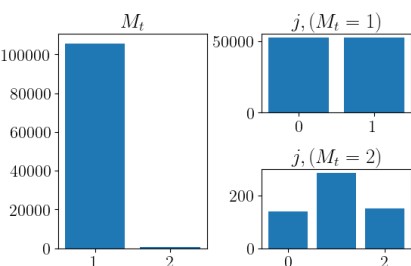

Figure 9: Distribution of number of judgements $M_t$, and resulting judgements $j$ for the CLIC data used for training.

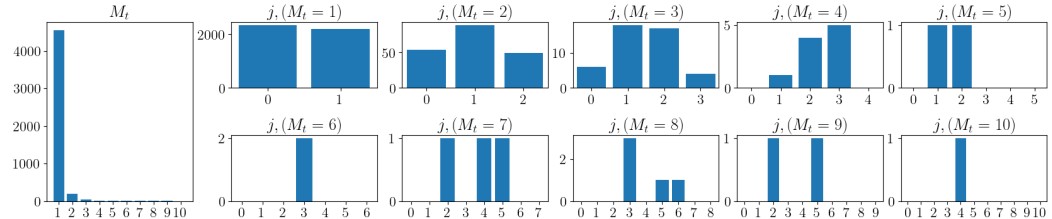

Figure 10: Distribution of number of judgements $M_t$, and resulting judgements $j$ for the CLIC data used for evaluation.

## D.2 Additional Visualisations

Fig 11 shows the distribution of triplets in the $(d_0, d_1)$ plane for the training set used to find $\hat{P}(d_0, d_1)$ from the CLIC dataset. Note that the triplets shown vary in number of judgements $M_t = \{1, 2\}$, and when training the triplets are treated as binary judgements ($M = 1$) on $M_t$ identical triplets.

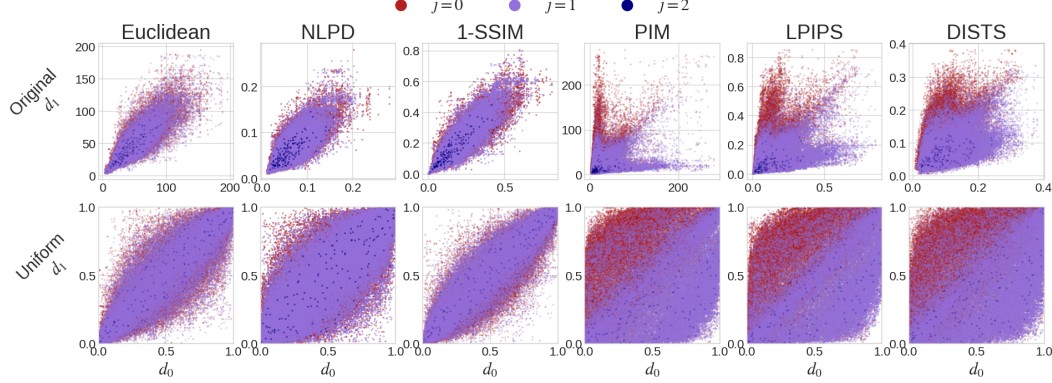

Figure 11: Candidate distances in their original space (top row) and uniformised (bottom row). Shown are the training samples from the CLIC dataset and the colour indicates the judgement assigned to the triplet according to $\{1, 2\}$ observers. The points in this plot have a varying number of observers $M$.

We also show the surface of the binomial parameter $\hat{P}(d_0, d_1)$ in the $(d_0, d_1)$ plane estimated from the CLIC training set.

## D.3 Evaluation on Training and Test set

Table 7 shows evaluation metrics on both the training and test set of CLIC. The training set has been used to fit $\hat{P}(d_0, d_1)$.

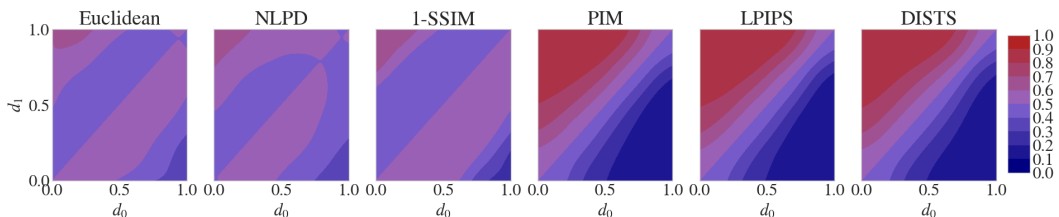

Figure 12: $\hat{P}(d_0, d_1)$ fit to the distribution of scores in the training set of CLIC, $\hat{P}(d_0, d_1)$, for different candidate distances.

Similar behaviour as the results on BAPPS can be observed in the negative log-likelihoods on the training set, where in some instances the negative log-likelihood of the actual measurements is lower than the theoretical minimum, again due to the number of samples we are using to estimate these properties. With a larger sample size, we expect this to not be an issue.

Table 7: Results on the CLIC dataset (Toderici et al., 2021). In the training set there are 2 judgements per triplet ($M = 2$) and in the test set, 1 ($M = 1$). Lower NLL is better.

| Measure | | Euclidean | NLPD | SSIM | PIM | LPIPS | DISTS |
|---|---|---|---|---|---|---|---|
| AJ$(n, \hat{P}, M)$ (%)↑ | Train | 53.185 | 51.860 | 56.315 | 73.034 | 73.226 | 72.082 |
| | Test | 44.116 | 45.415 | 44.872 | 74.023 | 74.016 | 75.991 |
| AJ$(\hat{n}, \hat{P}, M)$ (%)↑ | Train | 51.439 | 50.996 | 52.204 | 69.830 | 69.698 | 68.999 |
| | Test | 52.346 | 53.965 | 53.783 | 71.526 | 70.948 | 69.403 |
| NLL$(n, \hat{P}, M)$↓ | Train | 0.694 | 0.694 | 0.689 | 0.548 | 0.543 | 0.556 |
| | Test | 0.722 | 0.721 | 0.731 | 0.624 | 0.615 | 0.585 |
| NLL$(\hat{n}, \hat{P}, M)$↓ | Train | 0.694 | 0.695 | 0.693 | 0.567 | 0.567 | 0.582 |
| | Test | 0.718 | 0.717 | 0.713 | 0.586 | 0.588 | 0.609 |
| 2AFC Score↑ | Train | 0.5304 | 0.5173 | 0.5619 | 0.7297 | 0.7317 | 0.7202 |
| | Test | 0.4277 | 0.4389 | 0.4364 | 0.7318 | 0.7314 | 0.7539 |