# OpenReview forum: "Evaluating Perceptual Distances Models by Fitting Binomial Distributions to Two-Alternative Forced Choice Data"
_ICLR.cc/2025/Conference — Submitted to ICLR 2025_

### Official Review · Reviewer_dKVc · 2024-11-01

**Soundness:** 2
**Presentation:** 3
**Contribution:** 2
**Rating:** 5
**Confidence:** 4

**Summary:**

The paper presents a statistical model to improve the evaluation of perceptual distance models based on crowd-sourced 2AFC (two-alternative forced choice) data. The topic is interesting but there are some serious issues with the applicability and experiment.

**Strengths:**

1. Rigorous Statistical Framework: By employing binomial distribution and maximum likelihood, the approach brings statistical rigor to perceptual distance model evaluation.
2. Improved Insight into Judgments: The additional metrics offer nuanced insights into how perceptual models align with human judgments, capturing nuances beyond binary agreement.

**Weaknesses:**

**1. Applicability and Generalizability of the Proposed Metrics**

Could the evaluation metrics proposed in this study be extended to no-reference image quality assessment (NR-IQA) settings? In NR-IQA, pairwise comparisons, where participants judge the relative quality of two images, are common. This scenario bears similarities to 2AFC binary decision tasks and could benefit from the uncertainty modeling approach presented here. I recommend that the authors consider and discuss the metrics’ applicability in no-reference scenarios, potentially running experiments on pairwise NR-IQA datasets to further validate the transferability and generalizability of the metrics. This could significantly enhance the utility and impact of the proposed metrics in image quality assessment.

**2. Superiority Over Traditional Evaluation Metrics**

The study does not provide a clear demonstration of how the proposed metrics outperform traditional evaluation methods such as SRCC (Spearman’s Rank Correlation Coefficient), PLCC (Pearson Linear Correlation Coefficient), or KRCC (Kendall Rank Correlation Coefficient). While the authors show that deep learning models perform well using the new metrics, it is unclear whether this suffices to validate the effectiveness of the new metrics. To address this, I recommend a direct experimental comparison with these established metrics, which would highlight whether the new metrics provide additional information or improved accuracy. Moreover, it would be beneficial to discuss the potential of the new metrics to capture aspects like human judgment uncertainty and finer model differences, which traditional metrics may overlook, to enhance the credibility of this research.

**3. Dataset Selection and Applicability of Use Cases**

The experiments primarily rely on two datasets: BAPPS, which is diverse in distortion types, and CLIC, which is compression-specific. This dataset selection might limit the study’s ability to generalize the metrics’ effectiveness across different use cases. For instance, if the goal is to develop a general FR-IQA (full-reference image quality assessment) method, testing on a broader range of distortions—such as noise, blur, and color distortions—would be beneficial, possibly including datasets like TID2013 or LIVE. Conversely, if the focus is specifically on compression distortions, the paper would benefit from more explicitly discussing this application focus and how the proposed metrics could, for example, aid in choosing compression algorithms based on perceptual quality. Clearer examples or case studies illustrating how the metrics would assist in practical applications would improve the paper’s practical relevance.

**4. Relationship Between Uncertainty Modeling and MOS Distribution**

There seems to be a potential relationship between the proposed uncertainty modeling approach and MOS (Mean Opinion Score) distributions. MOS scores typically provide a more fine-grained rating (e.g., scores from 1 to 5), which offers a probabilistic view of perceived quality rather than binary decisions. I encourage the authors to explore how MOS distributions might contribute to refining the uncertainty modeling in this study, leveraging the probability distribution across the MOS scale to capture subtle perceptual differences more effectively. Discussing or experimenting with this connection could improve the interpretability and granularity of the proposed approach.

**Questions:**

Please see the weakness.

---

> ### Author Response · Authors · 2024-11-20
> **Reply to Reviewer dKVc Part 1**
>
> Thank you for your review and your comments.
>
> > Could the evaluation metrics proposed in this study be extended to no-reference image quality assessment (NR-IQA) settings? In NR-IQA, pairwise comparisons, where participants judge the relative quality of two images, are common. This scenario bears similarities to 2AFC binary decision tasks and could benefit from the uncertainty modeling approach presented here. I recommend that the authors consider and discuss the metrics’ applicability in no-reference scenarios, potentially running experiments on pairwise NR-IQA datasets to further validate the transferability and generalizability of the metrics. This could significantly enhance the utility and impact of the proposed metrics in image quality assessment.
>
> Thank you for this suggestion. This sounds like our proposed method could address a problem such as this, where the $(d_0, d_1)$ plane would simply be replaced with the NR-IQA for $x_1$ and $x_2$, and the rest of the method would remain the same. Could the reviewer suggest any no-reference image quality datasets, and their psychophysical counterparts, that could be used in such a way? We would like to include a brief discussion on this in the paper.
>
>
> > The study does not provide a clear demonstration of how the proposed metrics outperform traditional evaluation methods such as SRCC (Spearman’s Rank Correlation Coefficient), PLCC (Pearson Linear Correlation Coefficient), or KRCC (Kendall Rank Correlation Coefficient). While the authors show that deep learning models perform well using the new metrics, it is unclear whether this suffices to validate the effectiveness of the new metrics. To address this, I recommend a direct experimental comparison with these established metrics, which would highlight whether the new metrics provide additional information or improved accuracy. Moreover, it would be beneficial to discuss the potential of the new metrics to capture aspects like human judgment uncertainty and finer model differences, which traditional metrics may overlook, to enhance the credibility of this research.
>
> Correlation coefficients require the ranking of distorted images used in the dataset, which is not possible due to BAPPS dataset not training images between triplets and the CLIC dataset with limited overlap between triplets. We have added a sentence in Section 4.2 that makes it clear that this aspect of interpretability can be used to judge finer model differences on particular triplets, which traditional evaluation metrics do not address.
>
> > The experiments primarily rely on two datasets: BAPPS, which is diverse in distortion types, and CLIC, which is compression-specific. This dataset selection might limit the study’s ability to generalize the metrics’ effectiveness across different use cases. For instance, if the goal is to develop a general FR-IQA (full-reference image quality assessment) method, testing on a broader range of distortions—such as noise, blur, and color distortions—would be beneficial, possibly including datasets like TID2013 or LIVE. Conversely, if the focus is specifically on compression distortions, the paper would benefit from more explicitly discussing this application focus and how the proposed metrics could, for example, aid in choosing compression algorithms based on perceptual quality. Clearer examples or case studies illustrating how the metrics would assist in practical applications would improve the paper’s practical relevance.
>
> As stated in the paper, it is not possible to use datasets such as TID, LIVE, or CSIQ as these datasets release only the MOS value. Our method is designed for datasets gathered in a crowd-sourced-like approach, where practitioners are shown random triplets to provide judgements for. Datasets such as TID carefully select which triplets to show next, in a tournament-like fashion, to compute a mean opinion score. Additionally, computing MOS or a quality scale is not possible when triplets do not share distorted images. If the reviewer is aware of any other datasets that release the raw perceptual judgements rather than MOS, and where the triplets do not share images, please let us know.
>
> BAPPS contains a wide range of distortions, both traditional (noise, blur, color distortion) and deep learning model based distortions (super resolution, denoising) as well as composities of distortions. We feel in using both BAPPS and CLIC we cover traditional distortions, deep learning based distortions, and distortions coming from compression algorithms.

---

> ### Author Response · Authors · 2024-11-20
> **Reply to Reviewer dKVc Part 2**
>
> > There seems to be a potential relationship between the proposed uncertainty modeling approach and MOS (Mean Opinion Score) distributions. MOS scores typically provide a more fine-grained rating (e.g., scores from 1 to 5), which offers a probabilistic view of perceived quality rather than binary decisions. I encourage the authors to explore how MOS distributions might contribute to refining the uncertainty modeling in this study, leveraging the probability distribution across the MOS scale to capture subtle perceptual differences more effectively. Discussing or experimenting with this connection could improve the interpretability and granularity of the proposed approach.
>
> What the reviewer suggests is similar to the Thurstone approach and Maximum Likelihood Difference Scaling, where the quality score (or MOS) of a sample is modelled as a Gaussian distribution centered at the score, where the $\sigma$ of the distribution models the uncertainty in the score. This is a distinctly different issue than what we are trying to address; we do not wish to compute a quality scale but rather evaluate a distance using data collected from 2AFC experiments. Whilst MOS and other quality scores provide a ranking, we do not compute or evaluate the ranking but rather evaluate individual decisions. Additionally, our method is applied to datasets where computing MOS or any other quality score is infeasible since triplets do not share any distorted images.

---

> > ### Comment · Reviewer_dKVc · 2024-11-26
> > **Response to the authors**
> >
> > Thanks for the authors' response.
> >
> > Based on the comments of other reviewers. I would like to maintain my rating.

---

### Official Review · Reviewer_yquG · 2024-11-04

**Soundness:** 2
**Presentation:** 3
**Contribution:** 2
**Rating:** 3
**Confidence:** 5

**Summary:**

The paper presents an approach to evaluate perceptual distance models using two-alternative forced choice (2AFC) experiments, especially with crowd-sourced data where traditional ranking methods are not applicable. The authors use a binomial distribution to model the decision-making process, estimate parameters with maximum likelihood, and propose metrics that surpass simple accuracy measures. This method offers a more robust evaluation of perceptual models compared to classical and neural network approaches.

**Strengths:**

1. The paper introduces a novel method for evaluating perceptual distance models using binomial distributions to model decision-making in two-alternative forced choice (2AFC) experiments, which is an innovative attempt in the field of image quality assessment.
2. The paper demonstrates the robustness of the model to changes in hyperparameters and offers insights into the interpretability of the results.

**Weaknesses:**

1. The paper compares the proposed method with classical and neural network methods but does not extensively compare it with the latest state-of-the-art approaches in perceptual distance modeling.
2. The method is evaluated across different datasets, but there is insufficient evidence to demonstrate its generalizability across different types of images and distortions, particularly on traditional perceptual datasets like TID.
3. The experimental section may lack comparative experiments to show the specific improvements of the proposed method over existing ones.
4. The paper may suffer from a lack of clarity in certain sections, Improving the organization of the content can help address this issue.

**Questions:**

1. The authors should compare your method with state-of-the-art perceptual distance models.
2. It would be beneficial to have a more detailed and step-by-step explanation of the algorithm.

---

> ### Author Response · Authors · 2024-11-15
> **Reply to Reviewer yquG**
>
> We appreciate your review. Below are a few questions we have:
>
> >The paper compares the proposed method with classical and neural network methods but does not extensively compare it with the latest state-of-the-art approaches in perceptual distance modeling.
>
> Please could you provide references for the state-of-the-art approaches in perceptual distance modelling/evaluation? Note that we are not designing a distance, but rather modelling how well the distance is able to explain perceptual judgements from 2AFC experiments.
>
>
> > The method is evaluated across different datasets, but there is insufficient evidence to demonstrate its generalizability across different types of images and distortions, particularly on traditional perceptual datasets like TID.
>
> BAPPS contains many types of images, over 160,000 image patches from 5,000 images, and a large number of distortions, 425, including composite distortions. Datasets such as TID are well curated in controlled environments, however only report the mean opinion score (MOS), collected from using an elo rating in a tournament setting to decide which triplets to show observers. Additionally, triplets $(x_{ref}, x_0, x_1)$ share images which allows the authors to rank images and form the MOS. Our method is for evaluating perceptual distances using datasets where all triplets are independent (such as BAPPS), and a ranking cannot be made. Please could the reviewer provide references to datasets that follow the same methodology that our method was designed for.

---

### Official Review · Reviewer_Z66g · 2024-11-06

**Soundness:** 1
**Presentation:** 1
**Contribution:** 2
**Rating:** 3
**Confidence:** 5

**Summary:**

The paper proposes a probabilistic approach to model the subjective judgment obtained from 2AFC experiments about visual quality using a binomial distribution. The goal is to evaluate perceptual distance models predicting visual quality with respect to the 2AFC experimental data. Three metrics are derived for evaluation and the results on BAPPS are shown.

**Strengths:**

The issue of modeling 2AFC experimental data is an interesting topic.

The proposed method enjoys simplicity.

**Weaknesses:**

1. The paper is not easy to follow, mainly due to lack of definition, usage inconsistency, and unusual usage of notations and symbols. For instance, $t$ in $x_0(t)$ is not defined, $T$ is not defined, $d_v$ is not defined, $N$ is not defined, $P$ and $p$ are used without clear distinction, $j$ in Fig. 1 is not defined, the arguments are not consistent in $P$ (sometimes two arguments, sometimes three arguments, sometimes no argument, arguments do not match), {$(d_0, d_1), n=j$} doesn't make much sense, etc. In addition, some sentences are not easily understood, e.g., first sentence of Section 2.1, last sentence in page 6, what does 'preferred smoothness' mean and how can it be determined, etc.

2. Although the proposed method provides (maybe) new ways to evaluate the predicted quality scores by perceptual distance models, it is not experimentally shown whether the proposed method has any advantage over existing approaches including checking whether the image preferred by a majority of human raters is predicted to have a higher quality score by a model, comparing the predicted scores to the MOS estimated using the Thurstone or Bradley-Terry model on the subjective data, etc. In fact, in Table 2, the proposed method shows very similar results to the distance-only case, so the proposed method doesn't seem advantageous.

3. Section 4 presents experimental results, but they do not draw much meaningful conclusions. Above all, they do not prove that the proposed method is correct. Table 1 only shows that PIM, LPIPS, and DISTS are better than Euclidean, NLPD, and SSIM, which would be obvious without relying on the proposed method. Fig. 3 shows some difference among distance models, but not much more than that; it is said that humans perceive sharper differences when images are far from their reference, which I wouldn't agree. Fig. 4 is an interesting result, but it is not clear whether it can be useful practically. Fig. 5 shows the robustness of the proposed method against the choice of hyperparameters, but it does not prove that the results are correct.

4. It is said that the proposed method is free from assumptions unlike existing methods. However, the proposed method also imposes certain assumptions. For instance, it is assumed that the subjective preference strictly follows the binomial distribution without any noise, the probability in the binomial distribution depends only on the quality distances and nothing else, etc. Justification on these issues would be needed.

5. Even though the assumptions made in the proposed method are correct, the proposed model is fitted through maximum likelihood estimation, where fitting error wouldn't be zero. How accurate is the fitting in the experiments? The fitting error could be from from the inadequacy of the model, the multimodality of the likelihood, the violation of convexity of the optimization problem, or something else. Analysis on this would be needed to justify the adequacy of the proposed model.

6. Sampling for $\hat{n}$ is performed to obtain an 'upper bound' of AJ(?) and 'minimum possible' NLL. However, in Table 1, AJ with $\hat{n}$ doesn't act as an upper bound and NLL with $\hat{n}$ is not a lower bound. The meaning of the results of $\hat{n}$ should be interpreted appropriately. The same applies to Table 3.

(minor) Line 105, based off -> based on

**Questions:**

Please see the weaknesses.

---

> ### Author Response · Authors · 2024-11-20
> **Reply to Reviewer Z66g Part 1**
>
> Thank you for your review. Below we'll address some of the comments:
>
> > $t$ in $x_0(t)$ is not defined, $T$ is not defined, $d_v$ is not defined, $N$ is not defined, $P$ and $p$ are used without clear distinction, $j$ in Fig. 1 is not defined, the arguments are not consistent in $P$ (sometimes two arguments, sometimes three arguments, sometimes no arguments, arguments do not match), ${(d_0, d_1), n=j}$ doesn't make much sense, ect.
>
> $j$ is defined throughout the paper, as well as being defined in the caption of the figure (line 452). P is the binomial parameter, stated in line 197. $p$ is the empirical densities stated on line 246. The full notation for the binomial parameter is $P(d_0, d_1)$ but when referring to triplet $t$ we used $P(d_0(t), d_1(t))$. ${(d_0, d_1), n=j}$
> refers to the combinations of distances within triplets $(d_0, d_1)$ for which the number $n$ of people that preferred $x_0$ over $x_1$ was $j$, $j$ from 1 to $M$.
>  We have edited the beginning of Section 3 to make the notation and definitions clear.
>
> > In addition, some sentences are not easily understood, e.g., first sentence of Section 2.1, last sentence of page 6, what does 'preferred smoothness' mean and how can it be determined, ect.
>
> The sentence in section 2.1 concerns the different methodologies used in perceptual experiments. The last sentence of page 6 says that the M used to evaluate can be different from the one used in the estimation of $\hat{P}$ - we have rephrased this sentence. The 'preferred smoothness' is how smooth the estimation of $\hat{P}$ is, which depends on the $\sigma$ used in the kernel integration method. We have included this definition in the paper now.
>
> > Although the proposed method provides (maybe) new ways to evaluate the predicted quality scores by perceptual distance models, it is not experimentally shown whether the proposed method has any advantage over existing approaches including checking whether the image preferred by a majority of human raters is predicted to have a higher quality score by a model, comparing the predicted scores to the MOS estimated using the Thurstone or Bradley-Terry model on the subjective data, etc. In fact, in Table 2, the proposed method shows very similar results to the distance-only case, so the proposed method doesn't seem advantageous.
>
> We should clarify, that our model does *not* predict the quality score. We are evaluating the likelihood of the perceptual distance fitting experimental data from 2AFC experiments. This distinction is important. We evaluate whether the model conditioned on the perceptual distance predicts the same preferred image as the human evaluations, which is the 2AFC score in Eq. 9. We cannot compare to MOS estimated using the Thurstone or Bradley-Terry model, as stated in the paper, due to the triplets in the datasets having no images in common in the BAPPS dataset. If the reviewer is aware of any methods that work in such scenarios, we would be extremely interested in them. The results are similar to the distance-only case, however using the proposed method you can evaluate the likelihood of a given experimental result according to our model. This allows us to test certain scenarios in order to see where a particular perceptual distance fails, and answer questions such as: What happens if I have a few more judgements on this triplet? How likely is a judgement if $d_0$ is slightly reduced? How does the likelihood of a judgement change with a different perceptual distance?

---

> ### Author Response · Authors · 2024-11-20
> **Reply to Reviewer Z66g Part 2**
>
> > Section 4 presents experimental results, but they do not draw much meaningful conclusions. Above all, they do not prove that the proposed method is correct. Table 1 only shows that PIM, LPIPS, and DISTS are better than Euclidean, NLPD, and SSIM, which would be obvious without relying on the proposed method. Fig. 3 shows some difference among distance models, but not much more than that; it is said that humans perceive sharper differences when images are far from their reference, which I wouldn't agree. Fig. 4 is an interesting result, but it is not clear whether it can be useful practically. Fig. 5 shows the robustness of the proposed method against the choice of hyperparameters, but it does not prove that the results are correct.
>
> The correctness of the method is difficult to question. We have the maximum likelihood density $\hat{P}$, so by definition, the method is the most likely estimate according to the binomial distribution, which has been used throughout the literature when modelling human perceptual judgements. Fig. 3 provides interpretability for users evaluating their perceptual distance; you can see how the $\hat{P}$ changes with $(d_0, d_1)$. Fig. 4 adds to this and allows users to evaluate their perceptual distances with a different number of judgements, for example you could test how the log-likehood evolves as the triplet is evaluated by more people. Fig. 5 shows that the method is robust to hyperparameters, so rather than training a neural network (as in LPIPS), one could use our method with the default parameters to quickly evaluate their perceptual distance.
>
> > It is said that the proposed method is free from assumptions, unlike existing methods. However, the proposed method also imposes certain assumptions. For instance, it is assumed that the subjective preference strictly follows the binomial distribution without any noise, the probability in the binomial distribution depends only on the quality distances and nothing else, etc. Justification on these issues would be needed.
>
> We don't quite understand this criticism. We are modelling the noise as uncertainty in the judgement through the random choice in the binomial model with parameter $\hat{P}$. This is distinctly different than the Thurstone model where the *quality score* is seen as noisy. We are also evaluating whether the perceptual distances can explain the judgements, so if the binomial distribution depends on more than the perceptual distances, this will be seen in the results. We think these assumptions are straightforward and the justification is equally straightforward.
>
> > Even though the assumptions made in the proposed method are correct, the proposed model is fitted through maximum likelihood estimation, where fitting error wouldn't be zero. How accurate is the fitting in the experiments? The fitting error could be from from the inadequacy of the model, the multimodality of the likelihood, the violation of convexity of the optimization problem, or something else. Analysis on this would be needed to justify the adequacy of the proposed model.
>
> This is a little confusing, as the reviewer states the assumptions are correct, but the previous line criticises those assumptions.
> The fitting error is seen when evaluating the negative log-likelihood of the training data, and we see if this generalised to the test data. We can compare *how well* the models using different distances are fit using maximum likelihood. Appendix D.2 shows the conditional distributions used in the maximum likelihood estimation where they are not multimodal or complex. We do not see any benefit in the other analysis the reviewer is suggesting.
>
> > Sampling for $\hat{n}$ is performed to obtain an 'upper bound' of AJ(?) and 'minimum possible' NLL. However, in Table 1, AJ with $\hat{n}$ doesn't act as an upper bound and NLL with $\hat{n}$ is not a lower bound. The meaning of the results of $\hat{n}$ should be interpreted appropriately. The same applies to Table 3.
>
> We have clarified this in the text, that using observed judgements following the learned model provides a reference for comparison, rather than an upper bound.

---

> > ### Comment · Reviewer_Z66g · 2024-11-25
> >
> > I appreciate the authors' reply.
> >
> > However, major concerns, regarding correctness and advantages of the proposed method, still remain.
> >
> > The authors mention that some questions can be answered by the proposed method (at the end of the reply Part 1), but it is not quite clear what is the practical usefulness of answering such questions.
> >
> > If BAPPS is not suitable for making comparison to existing methods or for evaluating correctness, then maybe it is better to employ different (possibly new) datasets enabling appropriate evaluations. It is not clear why datasets like BAPPS (having no shared distorted images) should be used to evaluate perceptual distance models.

---

> > > ### Author Response · Authors · 2024-11-25
> > > **Reply to Reviewer Z66g**
> > >
> > > Thank you for your consideration.
> > >
> > > With regards to using BAPPS, this is a dataset that is used throughout the literature of proposing new perceptual distances [1,2 ,3]. We do not argue the correctness of doing so, we simple present a method that allows evaluation using BAPPS in a simple and efficient manner. Traditional perceptual datasets such as TID, CSIQ and LIVE contain measurements taken from controlled environments, and as such it is expensive and difficult to collect a large amount of data. BAPPS however drops this constraint, and crowd-sourced the perceptual judgements, allowing many more judgements on orders of magnitude more images. We do believe that this way of measuring perception (in an uncontrolled but data rich environment) will be used thoroughly in the future.
> > >
> > > [1] Ding, Keyan, et al. "Image quality assessment: Unifying structure and texture similarity." IEEE transactions on pattern analysis and machine intelligence 44.5 (2020): 2567-2581.
> > >
> > > [2] Severo, Daniel, Lucas Theis, and Johannes Ballé. "The Unreasonable Effectiveness of Linear Prediction as a Perceptual Metric." ICLR 2024.
> > >
> > > [3] Bhardwaj, Sangnie, et al. "An unsupervised information-theoretic perceptual quality metric." Advances in Neural Information Processing Systems 33 (2020): 13-24.

---

### Author Response · Authors · 2024-11-25
**Discussion**

We would appreciate if reviewers could take the time to read the responses to the reviews.

---

### Meta-Review · Area_Chair_uqsd · 2024-12-16

**Metareview:**

The paper was reviewed by three experts and there is an unanimity on the negative side with respect to the acceptance of this paper.

The responses provided by the authors did not convince the reviewers to improve their ratings.

The AC agrees with the reviewers that the paper is below the acceptance bar and invites the authors to benefit from the received feedback and further improve their work.

**Additional Comments On Reviewer Discussion:**

The responses provided by the authors did not convince the reviewers to improve their ratings.

Reviewer dKVc replies to the authors and maintains "marginally bellow acceptance rating", Reviewer Z66g appreciates the received response from the authors but at the same points out that major concerns remain and is "reject, not good enough", while Reviewer yquG rates "reject" and provides no reaction to the authors' response.

---

### Decision · Program_Chairs · 2025-01-22

Reject